# WoCoCo: Learning Whole-Body Humanoid Control with Sequential Contacts

**Chong Zhang**[†★]
ETH Zurich
chozhang@ethz.ch
[†] Equal Contribution
[★]Work was done at Carnegie Mellon University

**Wenli Xiao**[†]
Carnegie Mellon University
wxiao2@andrew.cmu.edu
[†] Equal Contribution

**Tairan He**
Carnegie Mellon University
tairanh@andrew.cmu.edu

**Guanya Shi**
Carnegie Mellon University
guanyas@andrew.cmu.edu

**Abstract:** Humanoid activities involving sequential contacts are crucial for complex robotic interactions and operations in the real world and are traditionally solved by model-based motion planning, which is time-consuming and often relies on simplified dynamics models. Although model-free reinforcement learning (RL) has become a powerful tool for versatile and robust whole-body humanoid control, it still requires tedious task-specific tuning and state machine design and suffers from long-horizon exploration issues in tasks involving contact sequences. In this work, we propose WoCoCo (Whole-Body Control with Sequential Contacts), a unified framework to learn whole-body humanoid control with sequential contacts by naturally decomposing the tasks into separate contact stages. Such decomposition facilitates simple and general policy learning pipelines through task-agnostic reward and sim-to-real designs, requiring only one or two task-related terms to be specified for each task. We demonstrated that end-to-end RL-based controllers trained with WoCoCo enable four challenging whole-body humanoid tasks involving diverse contact sequences in the real world without any motion priors: 1) versatile parkour jumping, 2) box loco-manipulation, 3) dynamic clap-and-tap dancing, and 4) cliffside climbing. We further show that WoCoCo is a general framework beyond humanoid by applying it in 22-DoF dinosaur robot loco-manipulation tasks. Website: lecar-lab.github.io/wococo/.

**Keywords:** Whole-Body Humanoid Control, Multi-Contact Control, Reinforcement Learning

## 1 Introduction

Humanoids are designed to operate in and interact with environments like humans do, which often requires the fulfillment of sequential contacts during task execution [1]. Provided specific contact plans, the typical solution is to employ model-based motion planning or trajectory optimization to generate whole-body references for tracking [2, 3, 4]. Although motion planning can be powerful for motion synthesis, it is often time-consuming and relies on simplified reduced-order dynamics models, which may affect the motion quality and the real-world performance [5, 6, 7, 8, 9, 10].

Model-free reinforcement learning (RL) has demonstrated remarkable robustness against model mismatch and uncertainties, and enabled real-time agile motions on legged robots [11, 12, 13, 14, 15]. However, these works focus on standard locomotion tasks (e.g., walking) without the necessity to fulfill specific contact sequences. Although some recent works have achieved RL-based locomotion with constrained footholds [16, 17, 18, 19, 20], they are heavily tuned for specific scenarios.

8th Conference on Robot Learning (CoRL 2024), Munich, Germany.

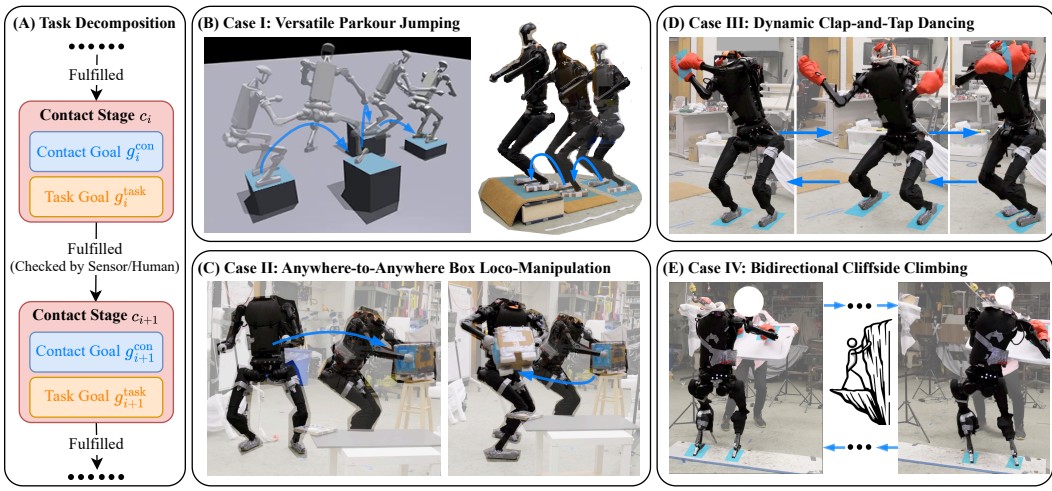

Figure 1: An overview of WoCoCo and tasks. (A) We decompose the task into separate contact stages, where each contact stage is defined by the contact goal and the task goal. (B)-(E): We applied our WoCoCo framework to various challenging tasks. Contact goals are visualized in blue, which involve some or all of the end effectors (i.e., hands and feet).

Similarly, in existing works that showcase task-aware contact sequences in real-world humanoids, such as soccer playing [21] and loco-manipulation [22, 23, 24], each RL policy is specifically tuned for a particular task or transition in the state machine. Distinct formulations and rewards are required for different policies, limiting their practical application in long-horizon tasks. Sferrazza et al. [25] trained policies for multiple dynamic manipulation tasks with a shared hierarchical RL architecture, yet they did not address sim-to-real concerns or propose unified contact-related rewards. Xiao et al. [26] enabled humanoids to track desired contacts with objects, but the controller was limited to animation purposes and required human motion references. To summarize, existing works on RL have shown potential in robustness and versatility, while a systematic and general method for controlling real-world humanoids under desired contact sequences using RL is missing.

In comparison, a single model-based solver such as Crocoddyl [5] can address multiple tasks with different contact sequences, requiring only slight adjustments to costs (few intuitive task-related terms). The question is: *How can we achieve such simplicity and adaptability with an RL framework?* Besides, regarding effective policy learning, we also identify three challenges: (1) Contacts are sparse, especially when coupled with other whole-body motion goals such as balancing and posture maintenance; (2) Robots may avoid exploring the whole long horizon due to the compounding risks; and (3) Sim-to-real transfer is non-trivial, and is often achieved through domain randomization [27] and task-irrelevant regularization rewards, which can hinder exploration.

In this work, we address the question by proposing WoCoCo, a general RL framework for whole-body humanoid control with sequential contacts. In WoCoCo, we reformulate the problem as the sequential fulfillment of multiple contact stages (detailed in Section 2), which also breaks down the exploration burden into separate stages. This then transforms each challenge to a question: **Q1**: How to reach desired contact states within each stage? **Q2**: How to streamline exploration across multiple contact stages? and **Q3**: How to develop a compatible sim-to-real pipeline?

We jointly tackle Q1 and Q2 by a concise yet effective WoCoCo reward design (detailed in Section 3.1), which is a combination of dense contact rewards, stage count rewards, and curiosity rewards. The dense contact rewards outperform standard 0-1 rewards [24] by counting every correct and incorrect contact, thereby guiding the policy more effectively. On top of that, the stage count rewards are proposed based on the number of fulfilled contact stages. This drives the robot to explore further stages to maximize cumulative rewards, thus mitigating the shortsightedness caused by the RL policy strategically staying in the current stage to avoid potential failures. To better facilitate exploration, we propose a task-agnostic curiosity reward term. Via detailed ablation analyses, we show that the WoCoCo reward design is both effective and minimal in Section 5.

In addressing Q3, we also propose a general sim-to-real pipeline with domain randomization and regularization rewards (Section 3.2). Inspired by [28], we design a curriculum with three training stages: initially training without domain randomization, then training with domain randomization, and finally increasing the weights of regularization rewards. This curriculum reduces the exploration burden introduced by sim-to-real modules in training. To summarize, our contributions are:

1. We propose WoCoCo, a general framework for RL-based whole-body humanoid control under sequential contact plans, with natural task decomposition based on contact stages.
2. We showcase how WoCoCo's task-agnostic designs empower end-to-end RL to tackle four challenging humanoid tasks and a 22-DoF dinosaur robot task, showing versatility and universality.
3. We validated the learned RL policies for the four aforementioned humanoid tasks in the real world, as shown in Fig. 1 and the videos. To our knowledge, these are the first instances of each task being solved by a single end-to-end RL policy.

## 2 Overview: Learning with Sequential Contacts and Task Decomposition

Considering a wide range of robotic tasks requiring active contacts for environment interactions, such as parkour jumping and loco-manipulation, we decompose these tasks into multiple contact stages $i \in \{0, 1, \ldots, I\}$ based on the desired contact sequences, as illustrated in Fig. 1. The robot is expected to sequentially fulfill these stages, where the fulfillment of each stage is defined as the simultaneous fulfillment of a contact goal $g_i^{\text{con}}$ (defining the contact states that certain end effectors should reach) and a task goal $g_i^{\text{task}}$ (defining additional task-specific requirements). In this paper, we study tasks where contact stages are predefined (e.g., heuristically designed), and our method can seamlessly be integrated with high-level contact planners (e.g., [29]). For example, in the parkour jumping task (Section 4.1 and Fig. 1(B)), each stepping stone corresponds to a contact stage where achieving correct foot contacts defines the contact goals, and maintaining upper body posture forms the task goals. Upon fulfilling a stage, checked by sensors or human observation, the robot advances to the next stage after a predefined arbitrary time period.

To develop RL-based controllers for these tasks, we formulate the policy learning problem as an extended Markov Decision Process (MDP) $\mathcal{M} = \langle \mathcal{S}, \mathcal{A}, \mathcal{T}, \mathcal{R}, \gamma, \mathcal{G}^{\text{con}}, \mathcal{G}^{\text{task}} \rangle$ of state $s_t \in \mathcal{S}$, action $a_t \in \mathcal{A}$, transition probability $\mathcal{T}$, reward $r_t \in \mathcal{R}$, discount factor $\gamma$, contact goal $g_i^{\text{con}} \in \mathcal{G}^{\text{con}}$, and task goal $g_i^{\text{task}} \in \mathcal{G}^{\text{task}}$. The objective is to maximize the expected return $\mathbb{E}\left[\sum_t \gamma^t r_t\right]$ by finding an optimal policy $a_t = \pi^*(s_t|g_{i:I}^{\text{con}}, g_{i:I}^{\text{task}})$. We define our rewards as

$$r = \underbrace{r_{\text{WoCoCo}} + r_{\text{reg}}}_{\text{task-agnostic}} + r_{\text{task}}, \tag{1}$$

where $r_{\text{WoCoCo}}$ is the task-agnostic rewards (detailed in Section 3.1), $r_{\text{reg}}$ is the task-agnostic regularization rewards for sim-to-real transfer (detailed in Section 3.2 and Appendix F), and $r_{\text{task}}$ is the task-related rewards with few intuitive terms (detailed in Section 4 for different tasks).

We employ Proximal Policy Optimization (PPO) [30] with symmetry augmentation [31] (detailed in Appendix D) to optimize the policy in Isaac Gym [32] simulation based on the parallel RL framework in [33]. All policies trained in this work are end-to-end MLP policies, while our framework does not restrict the policy architecture. The policy observations can include proprioception, exteroception (optional), and goal-related observations, which are detailed in Appendix H. The policy outputs $a_t$ are joint target positions tracked by low-level PD controllers to actuate the motors.

The remainder of the paper is organized as follows: In Section 3, we detail our task-agnostic $r_{\text{WoCoCo}}$ reward terms and sim-to-real designs. In Section 4, we show how our framework, WoCoCo, can be applied to a variety of challenging dynamic tasks with flexible definitions and representations of contact and task goals. We conduct further analyses and ablation studies in Section 5, and discuss the limitations and future works in Section 6.

## 3 WoCoCo Rewards and Sim-to-Real Transfer

This section presents our novel reward designs to overcome the challenges discussed in Section 1 and the sim-to-real pipeline.

### 3.1 WoCoCo Rewards

We propose WoCoCo rewards which comprise three task-agnostic terms:

$$r_{\text{WoCoCo}} = w_{\text{con}}r_{\text{con}} + w_{\text{stage}}r_{\text{stage}} + w_{\text{curi}}r_{\text{curi}}, \tag{2}$$

where $r_{\text{con}}$ is the contact rewards, $r_{\text{stage}}$ is the stage count rewards, and $r_{\text{curi}}$ is the curiosity rewards. As mentioned in Section 1, $r_{\text{con}}$ densifies the contact state reaching rewards, $r_{\text{stage}}$ incentivizes exploration across multiple contact stages, and $r_{\text{curi}}$ further facilitates exploration in the state space. These reward terms are detailed in the following subsections.

**Denser Contact Rewards: Every Contact Matters.** $r_{\text{con}}$ encourages correct contacts with task goal fulfillment, and offers denser rewards than 0-1 rewards[1] by additionally rewarding each correct contact while penalizing each wrong one. We define it as

$$r_{\text{con}} = n_{\text{corr}} - n_{\text{con}}n_{\text{wrong}} \cdot \mathbb{1}(n_{\text{stage}} > 0) + 2n_{\text{con}}^2 F_{\text{con}}F_{\text{task}}, \tag{3}$$

where $n_{\text{con}}$ is the maximal number of end effectors involved in the contact sequence, $n_{\text{corr}}$ and $n_{\text{wrong}}$ are respectively the number of end effectors with correct and wrong contacts at the current timestep, $n_{\text{stage}}$ is the number of fulfilled stages, $F_{\text{con}}$ and $F_{\text{task}}$ are respectively the boolean values for whether the contact goal or the task goal of the current stage is fulfilled. These symbols are further exemplified in Fig. 7 in Appendix A. The coefficients are designed to avoid local maxima, and the penalty for $n_{\text{wrong}}$ is masked during the first contact stage ($n_{\text{stage}} = 0$) to encourage exploration and avoid invalid episode reset (otherwise, the robot may immediately receive penalties upon reset).

**Stage Count Rewards: Do More, Get More.** Exploring new contact stages can come with failures and penalties, while staying at the current one may bring positive rewards. Hence, it is necessary to drive the agent towards new stages via rewards. To this end, we define the stage count reward as

$$r_{\text{stage}} = n_{\text{stage}}F_{\text{task}}, \tag{4}$$

and the condition $F_{\text{task}}$ is to avoid intentional unfulfillment of the task goal for cumulated stage count rewards.

**Curiosity Rewards: Drive the Exploration.** Curiosity rewards have been used to encourage exploraton in the RL context [34, 35]. For high-dimensional observations, random network distillation (RND) [36] has been proposed as a flexible and effective way for curiosity-driven exploration. In robot control, Schwarke et al. [24] have successfully applied RND to real-world whole-body loco-manipulation problems, while the curiosity observations are task-specific based on expert insights.

In this work, we aim to define curiosity rewards with task-agnostic observations that can be redundant and high-dimensional. We find that RND can over-explore states that do not generate meaningful behaviors, similar to what is reported in the RND work's Section 3.7 [36]. Instead, we propose to use count-based curiosity rewards via random neural network (NN) based hash, inspired by Tang et al. [37] and Charikar [38].

To be specific, we define a task-agnostic set of curiosity observations $o_{\text{curi}}$ (detailed in Appendix I). With a randomly initialized and frozen NN $f_{\text{curi}} : \mathbb{R}^{\dim(o_{\text{curi}})} \to \mathbb{R}^{\dim(\text{hash})}$, each $o_{\text{curi}}$ is hashed to a bucket with the index

$$\text{ID of bucket}(o_{\text{curi}}) = \text{BIN2DEC}\left[f_{\text{curi}}(o_{\text{curi}}) > \mathbf{0}\right], \tag{5}$$

where BIN2DEC interprets an array of boolean values as a binary number and converts it to the decimal format[2], and the curiosity reward is based on how many times the robot has visited states hashed in the same bucket:

$$r_{\text{curi}} = \frac{1}{\sqrt{\#. \text{ visits of bucket}(o_{\text{curi}})}}. \tag{6}$$

We find our curiosity rewards powerful and stable in facilitating exploration, even with task-agnostic curiosity observations across different challenging whole-body tasks. Based on the hash mechanism, overfitting of random networks can be constrained by the numerical decay as the #. visits increases.

---

[1]Reward 1 if the contact stage is fulfilled, else 0. Can be defined as $F_{\text{con}}F_{\text{task}}$ using the symbols in Eq. 3.

[2]For example, if $f_{\text{curi}}$ outputs $[1.5, -0.2, 0.4]$, the binary number is 101 and the bucket id is 5.

## 3.2 Sim-to-Real Transfer

Following existing works [39], we use domain randomization [27] and regularization rewards to enable sim-to-real transfer. The details are presented in Appendix. Notably, our domain randomization settings and regularization rewards are shared across all humanoid tasks.

We also apply a curriculum to reduce the exploration burden posed by domain randomization and regularization rewards, inspired by Li et al. [28]. Specifically, 1) we first train policies without domain randomization until they converge, 2) then resume training with domain randomization until convergence, and 3) afterwards increase the weights of regularization terms by 20% per 2000 iterations until they double, inducing more conservative behaviors. The curiosity rewards are activated only in 1). Following Li et al. [28, 40], we stack 3 control steps of previous joint states and actions, and append them to the policy observations to enhance the robustness by temporal memory.

## 4 Case Studies

In this section, we show how our framework, WoCoCo, can be applied to various challenging tasks with different contact sequences. As mentioned, we use the same $r_{\text{WoCoCo}}$ and $r_{\text{reg}}$ terms for different tasks, and the only task-specific adjustments are one or two very intuitive task rewards (introduced in each subsection). For brevity, we present the task definition, reward intuitions, and results here, detailing the observations and reward designs in Appendix.

### 4.1 Case I: Versatile Parkour Jumping

Parkour jumping by humanoids is a highly challenging dynamic task demonstrating advanced agility with precise landing, as showcased by Boston Dynamics (BD) [41]. However, BD's parkour motions come from a behavior library through offline trajectory optimization [42], which can limit versatility when deployed in the wild or when additional upper body motion is required for specific tasks. Li et al. [43] have achieved continuous bipedal jumping based on online model-based optimization, while only double-foot forward jumps without upper body tasks are supported. Li et al. [44] use RL to learn double-foot jumping in the 3D space, yet their method does not support continuous jumps, relies on a motion reference, and does not consider humanoids with upper body motions.

In contrast, we show that WoCoCo can enable end-to-end RL-based versatile parkour jumping with 1) single/double-foot contact switch, 2) controlled landing in the 3D space, and 3) upper body posture tracking, without any motion reference.

**Task Definition.** As shown in Fig. 2, we train the humanoid to jump over stones with various contact sequences, where each stone makes a contact stage. The contact goal is to have the correct foot (left/right/double) contact the stone, and the task goal is to maintain specified upper body postures ("hug"/"relax"). This setup challenges the robot to accurately execute foot contacts while adjusting its upper body posture during highly dynamic and coupled movements.

**Reward.** There is only one task-related reward term, encouraging tracking of the elbow position and orientation in the base frame to fulfill the task goal.

**Results.** The results are shown in Fig. 2, demonstrating the humanoid's capability to perform versatile continuous jumping while tracking upper body postures, and robustness against perturbations such as unseen gravels. In the real world, we only tested double-contact sequences with one or two continuous jumps due to facility constraints. Yet, the robot exhibited highly dynamic and adaptive behaviors for different stone heights and distances.

### 4.2 Case II: Anywhere-to-Anywhere Box Loco-Manipulation

Box loco-manipulation is an important application of humanoids and has been well studied with model-based controllers [45]. However, model mismatch and perturbations such as uneven terrains pose significant challenges to these controllers, for which RL can be a promising solution [13, 22].

That said, existing RL-based works either depend on finite state machines and train separate policies for each state transition with distinct formulations, rewards, and posture priors such as stance width [22, 23], or are limited to short-distance movements [24]. In this paper, we show that with

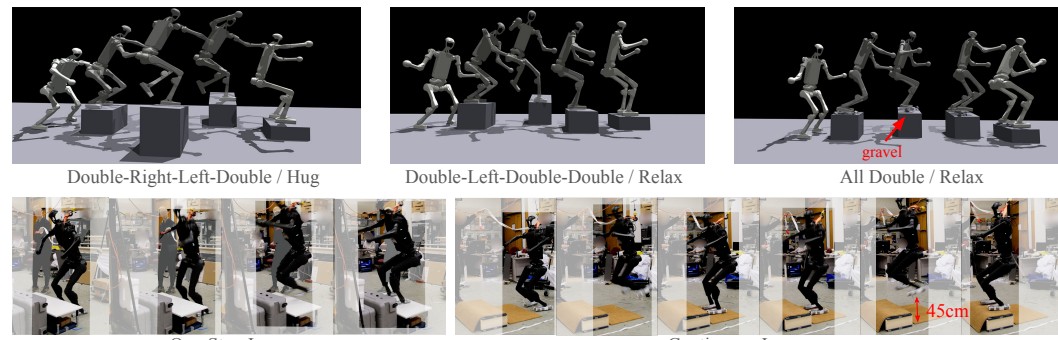

| One-Step Jump | Continuous Jump |

Figure 2: Learned versatile jumping motions in simulation and the real world. **Upper Row:** The humanoid performs continuous jumps with varying foot contact sequences and upper body posture goals, demonstrating robustness against unseen gravels. **Lower Row:** We transfer the policy to the real world, testing jumps with double-foot contacts at different heights and a "hug" posture.

provided current and goal positions[3] of the box, an end-to-end RL policy can control the humanoid to first approach the box and then transport it to the destination without any posture prior. The learned whole-body coordination can also enhance the motion efficiency, as observed in the existing works on quadrupeds [46, 47, 48, 49].

**Task Definition.** We define two contact stages. In the first stage, the contact goal is to place hands on both sides of the box, while the task goal is always fulfilled. In the second stage, the contact goal is to maintain hand contact with the box sides, and the task goal is to transport the box close to the destination. Placement at the destination is also feasible by modifying the contact goals to a virtual one. By defining the contact sequence solely on the hands, we leverage RL to achieve robust locomotion while simplifying the whole task.

**Reward.** There are two task-related reward terms, which incentivize minimizing the distances between the hands and the box, and between the box and its destination.

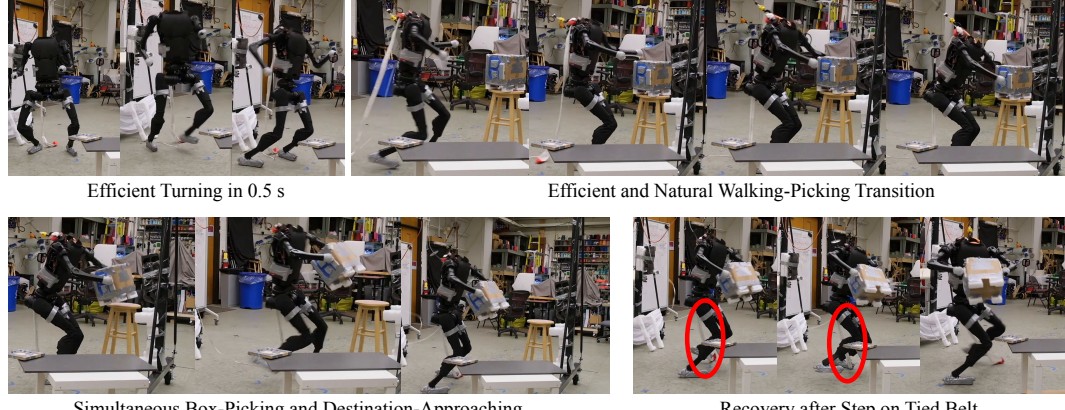

| Efficient Turning in 0.5 s | Efficient and Natural Walking-Picking Transition |
| Simultaneous Box-Picking and Destination-Approaching | Recovery after Step on Tied Belt |

Figure 3: Learned whole-body box loco-manipulation behaviors in the real world.

**Results.** As shown in Fig. 3, the humanoid can efficiently turn, transition seamlessly between walking and picking, and simultaneously approach the destination while picking up the box. It can also recover after stepping on a belt tied to itself, showcasing robustness.

### 4.3 Case III: Dynamic Clap-and-Tap Dancing

Humanoids may also entertain with dynamic dancing skills. BD has achieved impressive dancing with model-based control and offline trajectory optimization [50]. Existing RL-based methods can track human references [39, 51], yet unable to ensure accurate tapping on the ground. Here we show WoCoCo can enable RL-based dynamic dancing with accurate tapping and optional clapping.

---

[3]Referred to as "destination" to avoid confusion with contact/task goals.

**Task Definition.** In this task, contact stages are assigned to feet and hands. As shown in Fig. 4, there are three moves to compose diverse contact sequences where "Left" and "Middle" can transit to each other and so do "Right" and "Middle". In each contact stage, the task goal is to position the hands within the black bounding boxes (predefined in the base frame). The contact goal requires foot contact with the ground in their corresponding bounding boxes (predefined in the world frame), plus hand self-collision if the move is "Left" or "Right".

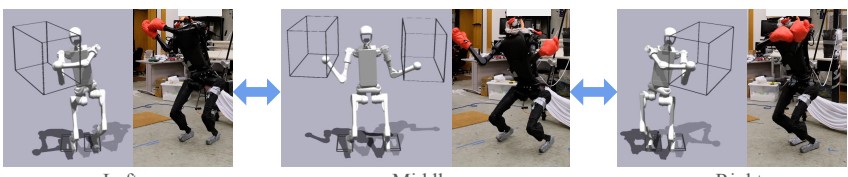

Left           Middle           Right

Figure 4: Learned dancing motions in simulation and the real-world. Black bounding boxes indicate the foot contact goals and the hand task goals.

**Reward.** There are two task-related rewards, one encourageing spreading the arms, and the other incentivizing minimizing the distances between the feet and the centers of their goal contact regions.

**Results.** We successfully learned the policy with real-world deployment, as shown in Fig. 4.

### 4.4 Case IV: Bidirectional Cliffside Climbing

Cliffside climbing is a representative task requiring precise movement of all limbs to support the humanoid. Though model-based controllers [9, 52, 53, 54] have showcased success in such problems, we prove RL is also a promising solution for fast and resilient multi-contact locomotion.

**Task Definition.** In this task, the contact sequences are tracked to make the humanoid move along the cliffside, as shown in Fig. 5. In each contact stage, the task goal is always fulfilled. The contact goal requires both hands to touch the goal regions on the wall, while both feet need to stand on their goal regions on the ground. Each end effector's goal region is bounded by a 2-d square.

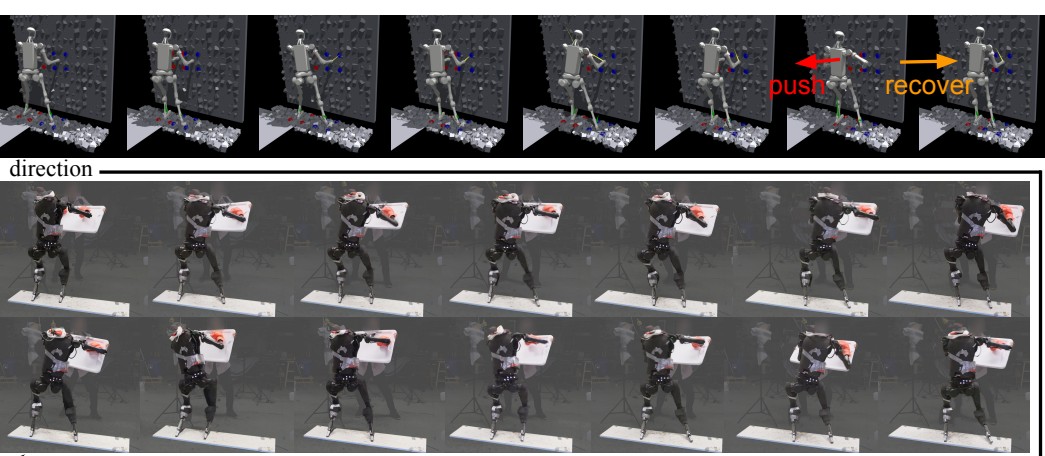

Figure 5: Learned cliffside climbing behavior in simulation and the real-world. The humanoid exhibited resilience against perturbations and compliance during contact with unseen gravels.

**Reward.** There are two task-related reward terms. The first encourages the humanoid to face the wall, and the second incentivizes precise movement of end effectors by minimizing the distances between the end effectors and the centers of their goal contact regions.

**Results.** The learned cliffside climbing behavior is shown in Fig. 5. The policy is robust against pushes and unseen gravels in simulation. In the real world, the cliff is replaced by a board held by a human, and the humanoid can adapt to varying contact forces on the hand during the interaction.

### 4.5 Beyond Humanoid: Dinosaur Loco-Manipulation

To show WoCoCo can generalize to other embodiments, we trained a 22-DoF dinosaur robot (adapted from [55]) to perform a ball loco-manipulation task. This involves pushing a ball to a

specified destination using one of its six end effectors (head, tail, and four feet). The task definition mirrors that of box loco-manipulation, except that the desired contact point is the projection of the destination through the ball's center to its surface, facilitating the ball's movement toward the destination. Contacts made by the end effector with the ball near this point fulfill the contact goal. The results are shown in Fig. 6.

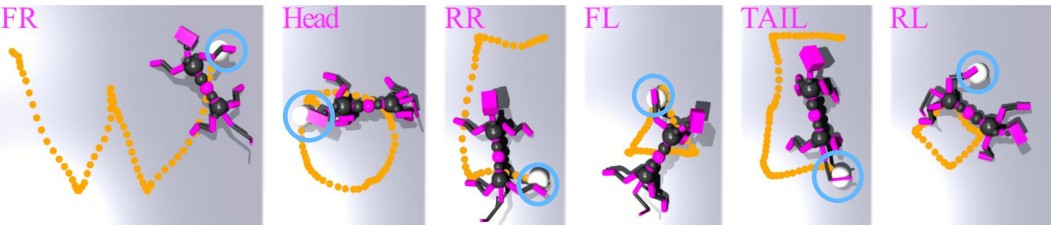

Figure 6: We train the dinosaur robot to push the ball towards destinations with different end effectors. By altering the destinations, we make the robot generate ball trajectories forming "WoCoCo".

## 5  Analyses and Ablations

Given that versatile parkour jumping is arguably the most challenging task, and training every task for the same setting is costly, we do analyses and ablation studies based on the jumping task. The learned behaviors of baselines are visualized in Fig. 8 in Appendix B.

**Ablating Dense Contact Rewards.** With 0-1 contact rewards $r_{\text{con}}^{0-1} = c^{0-1} F_{\text{con}} F_{\text{task}}$, the humanoid cannot explore to jump over the stones, and tracks upper body postures without moving. This proves the necessity of our dense contact rewards.

**Ablating Stage Count Rewards.** Without the stage count rewards, the humanoid intentionally does not fulfill the contact goal to avoid progressing to further contact stages, while still obtaining other rewards. This verifies the effectiveness of our proposed stage count rewards.

**Ablating Curiosity Rewards.** Without the curiosity rewards, the humanoid cannot jump over the stones, and tracks upper body postures without moving, which means under-exploration. With RND-based curiosity rewards, the humanoid learns to lean backward in a risky way, which aligns with the observations by Burda et al. [36]: the agent may over-explore a dangerous behavior pattern while staying alive, as such states are rare in the agent's experience compared to safer ones. In comparison, our curiosity rewards achieves effective exploration without overfitting specific behaviors.

**Empirical Benefits of WoCoCo.** With WoCoCo, the humanoid demonstrates high agility and motion efficiency. These motions are not constrained by simplified models and motion priors. Besides, by training with diverse task configurations, the learned RL policies can fulfill versatile contact goals. The policies also showcase robustness against perturbations such as unseen gravels.

**Training Stability.**  Despite the stochasticity of curiosity-driven exploration as shown in [24], our method has been stable against random network initialization and exploration. This is shown by Fig. 10 in Appendix C where we plot the learning curves for five different random seeds.

## 6  Limitation and Future Works

One limitation of our work is the lacking knowledge of when the controller will fail. In contrast, model-based methods can explicitly tell whether they can find a feasible solution. Therefore, we may explore failure predictors [56] and other safety assessment methods in the future [57]. Besides, if the contact sequence length is unknown a priori, we may need heuristic reward clamping to avoid the robot exploiting the stage count reward.

We currently rely on motion capture as a prototype, and we will try to incorporate onboard sensing in the future. We will also explore sampling-based [53] or LLM-based [26] high-level planners, while currently we predefine contact sequences based on heuristics. Another limitation is the requirement for explicit contact feedback (by contact sensors or human observers) to switch contact stages, a process that might be implicitly managed by the policy in the future.

**Acknowledgments**

We thank Arthur Allshire and Jason Liu's help with simulation, Milad Shafiee's help with the dinosaur robot, Guanqi He's help with the hardware, Justin Macey and Jessica Hodgins' support of facilities. We thank Boston Dynamics for inspiring us with awesome works.

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

# Appendix

## A    An Illustrative Example of Symbols in Contact Rewards

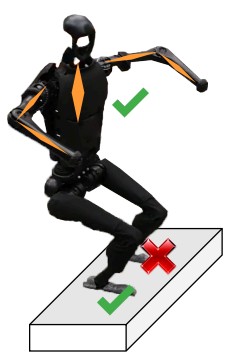

Here we exemplify the contact reward calculation with the parkour jumping case. At the timestep illustrated in Fig. 7, the contact goal is to have the right foot on the stone with the left foot in the air, and the task goal is to maintain a "hug" posture with the upper body. We have:

- Contact sequence is defined on the feet: $n_{con} = 2$.
- Task goal fulfilled, $F_{task} = 1$.
- Contact goal not fulfilled, $F_{con} = 0$.
- Number of correct contacts: $n_{corr} = 1$ (right foot).
- Number of wrong contacts: $n_{wrong} = 1$ (left foot).

Supplementary Figure 7: Illustration of symbols, explained on the right.

Then, based on Equation (3), the contact reward at the current timestep is:

$$r_{con}^{example} = 1 - 2 \cdot \mathbb{1}(n_{stage} > 0) + 0, \tag{7}$$

which means 1 if it is in the first contact stage, and $-1$ otherwise.

## B    Ablation Baseline Behaviors

We visualize the learned behaviors of baselines in ablation studies in Fig. 8. We also show their corresponding learning curves below for the average curiosity values and task progress (i.e., the number of fulfilled contact stages divided by its maximum number).

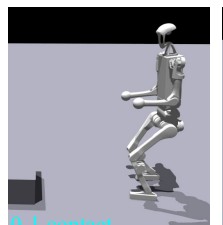 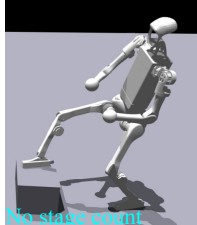 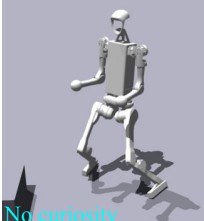 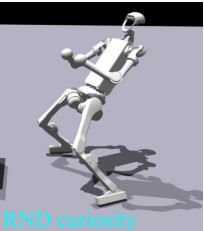

Supplementary Figure 8: Learned behaviors of baselines. 0-1 contact rewards: failed exploration. No stage count rewards: intentional non-fulfillment of the contact goal. No curiosity rewards: failed exploration. RND-based curiosity rewards: leaning backward in a risky way.

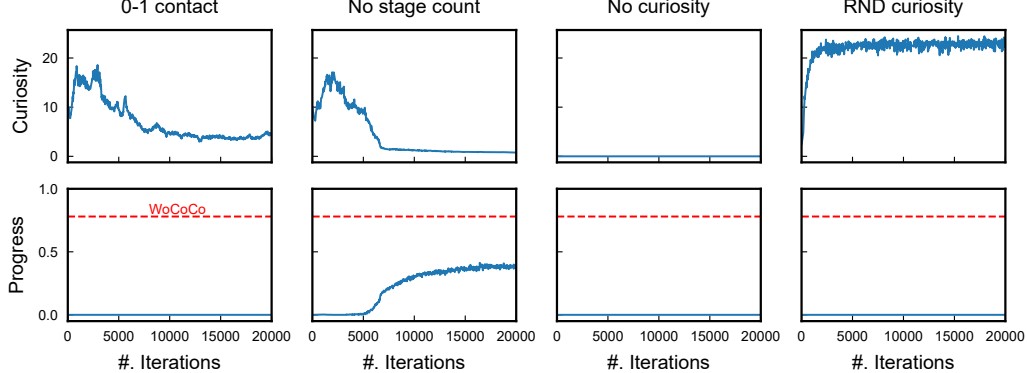

Supplementary Figure 9: Corresponding learning curves of baselines. The convergent average progress of WoCoCo is also visualized in red, marking successful policy learning.

## C    Training Stability

The average curiosity values and task progress (i.e., the number of fulfilled contact stages divided by its maximum number) are presented in Fig. 10 for the versatile parkour jumping task.

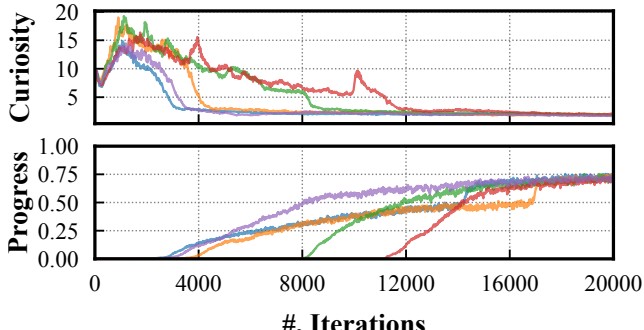

Supplementary Figure 10: Learning curves of five random seeds for parkour jumping learning.

## D    Symmetry Augmentation

Following Hoeller et al. [15] and Zhang et al. [18], to improve data efficiency, we leverage symmetry augmentation [31] in the PPO algorithm based on the humanoid's left-right symmetry.

To enforce symmetry in the curiosity rewards, we compute $r_{\mathrm{curi}}$ of both the original curiosity observation $o_{\mathrm{curi}}$ and its augmented observation, and take their average value.

## E    WoCoCo Reward Weights

Supplementary Table 1: WoCoCo Reward Weights

| Term | Parkour | Loco-Mani. | Dancing | Cliffside |
|---|---|---|---|---|
| $w_{\mathrm{con}}$ | 120 | 40 | 10 | 20 |
| $w_{\mathrm{stage}}$ | 160 | 160 | 5 | 40 |
| $w_{\mathrm{curi}}$ | 20000 | 40000 | 5000 | 10000 |

## F    Regularization Rewards

Here we use $v$ for linear velocities, $\omega$ for angular velocities, $q$ for joint positions, $\tau$ for joint torques, $a$ for joint actions, $F_c$ for contact forces, and $\theta_c$ for the contact angles (0 if normal or no contact). If jumping motions are not desired (i.e., all other tasks except parkour jumping), two style shaping terms ("feet air time" and "no fly") are needed.

Supplementary Table 2: Regularization Rewards

| Term | Expression | Weight |
|------|------------|--------|
| Yaw rate | $\omega_z^2$ | $-0.1$ |
| Torques | $\sum_{\text{joint}} \left(\frac{\tau}{\tau_{\text{lim}}}\right)^2$ | $-0.5$ |
| Torque overlimit | $\sum_{\text{joint}} \max\left(\frac{|\tau|}{\tau_{\text{lim}}} - 0.95, 0\right)$ | $-500$ |
| DoF acceleration | $\sum_{\text{joint}} \ddot{q}^2$ | $-0.000005$ |
| DoF velocities | $\sum_{\text{joint}} \dot{q}^2$ | $-0.003$ |
| Action rate | $\sum_{\text{joint}} \dot{a}^2$ | $-250$ |
| Termination | $\mathbb{1}(\text{termination})$ | $-200$ |
| Foot contact forces | $\sum_{\text{foot}} \max(|F_c| - 550, 0)$ | $-0.005$ |
| Foot orientation | $\sum_{\text{foot}} |\sin \theta_c|$ | $-50$ |
| Stumble | $\sum_{\text{foot}} \mathbb{1}(\text{horizontal impact})$ | $-100$ |
| Slippage | $\sum_{\text{foot}} |v|^2 \cdot \mathbb{1}(\text{contact})$ | $-5$ |
| *if jumping not desired* | | |
| Feet air time | $T_{\text{air}} - 0.5$ [33] | $20$ |
| No fly | $\mathbb{1}(\text{foot contact})$ | $10$ |

# G    Task-Related Rewards

## G.1    Versatile Parkour Jumping

Only one task-related reward is introduced for parkour jumping:

$$r_{\text{task}} = w_{\text{task}} \exp(-\frac{|\text{err}_{\text{rot}}|}{\pi}) \exp(-\frac{|\text{err}_{\text{pos}}|}{1}), \tag{8}$$

where $|\text{err}_{\text{rot}}|$ and $|\text{err}_{\text{pos}}|$ are respectively the rotation and position errors of the elbow w.r.t. the desired orientations and positions in the base frame. We have $w_{\text{task}} = 30$.

## G.2    Anywhere-to-Anywhere Box Loco-Manipulation

The task rewards include two terms:

$$
\begin{aligned}
r_{\text{task}} = \quad & w_{\text{box}} \exp(-d_{\text{box2dest}}) \cdot \mathbb{1}(n_{\text{stage}} > 0) \cdot \mathbb{1}(|\theta_{\text{dest}}| < \frac{\pi}{2}) \\
+ & w_{\text{hand}} \exp(-\frac{d_{\text{left2box}} + d_{\text{right2box}}}{2}) \cdot \mathbb{1}(|\theta_{\text{box}}| < \frac{\pi}{6}),
\end{aligned}
\tag{9}
$$

where $d_{\text{box2dest}}$ is the distance between the box and the destination, $d_{\text{left2box}}$ and $d_{\text{left2box}}$ are respectively the distances from the left hand and the right hand to the corresponding box side center, $\theta_{\text{dest}}$ and $\theta_{\text{box}}$ are respectively the direction angles of the destination and the box in the base frame. These two terms encourage approaching and moving the box. We have $w_{\text{hand}} = 100, w_{\text{box}} = 200$.

For dinosaur ball loco-manipulation, we remove the direction angle conditions.

## G.3    Dynamic Clap-and-Tap Dancing

The task rewards include two terms:

$$
\begin{aligned}
r_{\text{task}} = \quad & w_{\text{hand}} \left({}_b y_{lh} I_{lh} + {}_b y_{rh} I_{rh}\right) \\
+ & w_{\text{foot}} \exp\left(-(\frac{d_{\text{lf2box}}}{0.1})^2\right) \exp\left(-(\frac{d_{\text{rf2box}}}{0.1})^2\right),
\end{aligned}
\tag{10}
$$

where ${}_b y_{lh}$ and ${}_b y_{rh}$ are respectively the y values (displacement from the sagittal plane) of the left hand and the right hand in the base frame, $I_{lh}$ and $I_{rh}$ are respectively the signs of the $y$ values for the centers of the left and right hand bounding boxes in the base frame, $d_{\text{lf2box}}$ and $d_{\text{rf2box}}$ are respectively the distances from the left foot and right foot to the centers of their bounding boxes. These two terms encourage arm spread with precise footholds. We have $w_{\text{hand}} = 5, w_{\text{foot}} = 10$.

### G.4 Bidirectional Cliffside Climbing

The task reward includes two terms:

$$r_{\text{task}} = w_{\text{base}}(-|\theta_{\text{wall}}|) + w_{\text{ee}} \sum_{j=1}^{4} \exp\left(-\frac{d_{\text{ee2goal,j}}}{0.2}\right), \tag{11}$$

where $\theta_{\text{wall}}$ is the angle of the wall relative to the base yaw, and $d_{\text{ee2goal}}$ is the distances from the four end effectors to the centers of their goal regions. These two terms encourage the humanoid to face the wall while maintaining precise end effector placement. We have $w_{\text{base}} = 50, w_{\text{ee}} = 5$.

## H  Policy Observations and Architechtures

The policy observations in this work consist of two parts for all policies: proprioception and goal representations. Camera and LiDAR observations are not used here but our framework does not exclude them. The goal representations are provided by the motion capture system as a prototype.

### H.1  Proprioception

We use the following proprioception observations (shared across all tasks): joint positions, joint velocities, previous actions, base linear and angular velocities, and projected gravity. Joint positions, joint velocities, and previous actions are stacked by 3 control steps.

### H.2  Goal Representations

For versatile parkour jumping, we feed the future two stages' goal representations to the policy network, so the robot can adapt the foothold for future goals. For other tasks, we feed only the current stage's goal representations. This design allows our policies to generalize to varying sequence lengths during deployment.

#### H.2.1  Versatile Parkour Jumping

The contact goal is represented by the corner points of each foot's next two stones in the base frame, which are set to zeros when the foot is intended to be in the air. The task goal is represented by the desired elbow orientation and position in the base frame.

#### H.2.2  Anywhere-to-Anywhere Box Loco-Manipulation

The contact goal representation is the center points of the box sides in robot's base frame, and the task goal representation is the destination position in the base frame.

#### H.2.3  Dynamic Clap-and-Tap Dancing

We use a combined contact-and-task observation: the one-hot vector for the case (Left or Middle or Right) plus the robot's $x$, $y$, and yaw values in the world frame. This accommodates scenarios where the humanoid is required to dance within a fixed area.

#### H.2.4  Bidirectional Cliffside Climbing

The task goal is always fulfilled so we only need the contact goal representation: corner positions in the base frame for all the goal regions of the current and next contact stages.

### H.3  NN Architechtures

Actors and Critics are all MLPs with $[512, 256, 128]$ hidden units.

# I   Curiosity Details

## I.1   Curiosity Observations

Our curiosity observations comprise the robot's base states and end effector states. The base states comprise the world-frame position, orientation (quaternion), linear and angular velocities of the base. The end-effector states comprise the world-frame positions and contact states of all the end effectors.

For numerical stability and generalization, We considered the following 3 methods of preprocessing:

  (1) Normalizing the observations into $[0, 1]$ based on their maximal possible ranges.
  (2) Normalizing the observations into $[0, \pi]$ based on their maximal possible ranges, and convert each value to its $\sin$ and $\cos$ values.
  (3) On top of (2), rescaling these values with $n_{\text{stage}} + 1$ so the curiosity becomes stage-aware.

We found no significant difference in the outcomes for the above settings, and the results reported in this paper are with (2). However, we present all these settings here to inspire further research and discussion.

## I.2   NN Architechtures

Curiosity Hash networks have one hidden layer with 32 hidden units, and 16-d outputs.

# J   Domain Randomization

Supplementary Table 3: Domain Randomization for Sim-to-Real Transfer

| Term | Value |
|---|---|
| Friction | $\mathcal{U}(0.2, 1.1)$ |
| Base CoM offset | $\mathcal{U}(-0.1, 0.1)$m |
| Link mass | $\mathcal{U}(0.7, 1.3) \times$ default kg |
| P Gain | $\mathcal{U}(0.75, 1.25) \times$ default |
| D Gain | $\mathcal{U}(0.75, 1.25) \times$ default |
| Torque RFI [58] | $0.1 \times$ torque limit N $\cdot$ m |
| Control delay | $\mathcal{U}(0, 20)$ms |
| Push robot | interval $= 5s$, $\Delta v_{xy} = 0.25$m/s |
| Terrain type | flat / unseen gravels |

# K   Significance of Sim-to-Real Curriculum

Our training curriculum for sim-to-real transfer has 3 phases. Here we show the significance of such design by ablations.

## K.1   Moving Domain Randomization to Phase 1

If we move the domain randomization from phase 2 to phase 1, i.e., to merge the first two phases, we find the policy learning can be hindered by heavy randomization. For example, when learning the parkour jumping policy, this merge leads to the average progress converging below $0.6$, compared to the ones $> 0.75$ in standard WoCoCo.

## K.2   Canceling Phase 3

In phase 3, we graudally increase the weights of the regularization terms. Without this phase, we find the learned behaviors can be aggressive in the real world, which may lead to failures even for the least dynamic task, i.e., cliffside climbing, as shown in Fig. 11.

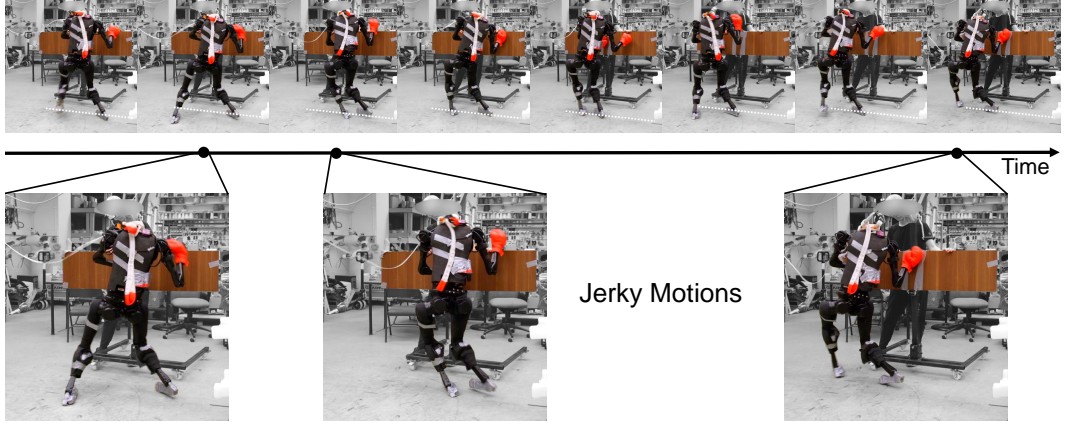

Supplementary Figure 11: Without the last phase of the sim-to-real curriculum, the real-world behaviors can be jerky. For example, during cliffside climbing, the robot often stepped out of the cliff edge (indicated by white dotted lines) due to jerky motions, while all of the contact goals were set within the boundary.

## L  Deployment Details

### L.1  Control

Our policy updates at 50 Hz, and the PD controller updates torque commands at 200 Hz. We apply a Butterworth low-pass filter of 4-Hz bandwidth to the PD targets to avoid jerky outputs.

### L.2  Real-to-Sim

We found the official URDF file contains significantly biased torso mass and wrong foot geometry, so we weighed the torso mass and measured the foot geometry by our own.

