# OpenReview forum: "WoCoCo: Learning Whole-Body Humanoid Control with Sequential Contacts"
_robot-learning.org/CoRL/2024/Conference — CoRL 2024_

### Official Review · Reviewer_MQ1w · 2024-07-11
**The significance of the paper is considered strong.**

**Originality:** 3
**Technical Quality:** 3
**Clarity Of Presentation:** 2
**Potential Impact:** 3
**Recommendation:** 3
**Confidence:** 3

**Review:**

Strengths:
* This paper is significant because it shows that the proposed framework can generate a variety of motions in real humanoid robots.
* The accompanying video and the video available on the project web page are very impressive.

Weaknesses:
* The abstract and introduction need to be improved. For example, the authors criticize conventional RL for requiring careful design of state machines and distinct rewards, but WoCoCo also requires decomposition of tasks into contact stages and design of task-specific reward functions. It is a subjective to say that the decomposition into contact stages and the design of the reward function in WoCoCo are simpler, so the improvement from the conventional RL is a bit unclear. Furthermore, it is not clear why (1)-(3) are the primary challenges to the question "How can we achieve such simplicity and adaptability with an RL framework? It is also not clear why challenges (1)-(3) turns into the three key questions Q1-Q3.
* Lack of experimental validation makes it difficult to admit that the development of the training curriculum is a contribution of this paper. Thus, the methodological contribution of this paper appears to be weak, as it only proposes the design of an effective reward function. I would like to see the effectiveness of the training curriculum if it could be easily demonstrated, e.g., through an ablation study.
* It would be easier to understand the effectiveness of the proposed reward function if the learning curves for the ablation study in Section 5 and Appendix B were also shown.
* I think it would be better to delete the DANCING WITH SKULLS quote on page 4, as it is not clear to readers who have not read the RND work [36] and would make this paper less self-contained.
* I recommend that the authors describe the details of real robot control. For example, the following details would be useful
(i) Have system identifications of robots and actuators been performed to reduce the sim-to-real gap?
(ii) At what hertz are the policies updated?
(iii) At what hertz is the low-level PD controlled?
(iv) Is a low-pass filter applied to the policy?
* Typo on page 8: Besides, By training with ...

**Quality Of The Limitations Section:**

3

**Questions For Rebuttal:**

Issues that need to be addressed by the authors are listed in Weaknesses in the Review section.

**Robotics Focus:**

4

**Summary Of Paper:**

This paper proposes WoCoCo, a whole-body control framework for humanoid robots using reinforcement learning. In WoCoCo, a task was decomposed into several contact stages, and policies were learned such that the contact and task goals at each stage were satisfied. To enable the transfer of policies to a real robot, WoCoCo learned the policies through a training curriculum using domain randomization and regularization rewards.

**Summary Of Recommendation:**

The contribution of this paper is sufficient: strength in significance of the paper. The experiments are performed in not only simulation but also real environments.

---

### Official Review · Reviewer_HtJA · 2024-07-12
**Valuable insight in using contact sequence as a means of conveying motion command to humanoid policy.**

**Originality:** 4
**Technical Quality:** 3
**Clarity Of Presentation:** 2
**Potential Impact:** 3
**Recommendation:** 3
**Confidence:** 5

**Review:**

This paper describes a valuable formulation of using contact sequences to convey motion commands to the humanoid policy. This framework could potentially impact the general humanoid algorithm implementation. The author provides a clear and well-organized description of the motivation, proposal, and experiment results. However, it is hard to picture each case study's implementation and the numerical formulation. The technical details could be improved.

**strength**
* The proposal of using contact as a high-level command representation is quite a universal and important direction to the humanoid low-level control.

* This work demonstrates the importance of using contact as a high-level command with sufficient case studies.

* The insight of adding correct and wrong contacts into reward formulation is valuable.

**weakness**
* The title of Section 3 *How to train your Droid* does not sound academic. It would be better to rephrase it.

* The author fails to explain how the contact points are assigned in the real world, especially in the case of *Versatile Parkour Jumping*.

**Quality Of The Limitations Section:**

3

**Questions For Rebuttal:**

1. Is all the contact goal $g_i^\text{con}$ and task goal $g_i^\text{task}$ fed into the policy network in each timestep? If not, how does the author select the goal to feed into the policy network?

2. How does the author specify the goal positions in the robot base frame in the real world, since the author claims that they do not use exteroception? For example the platform position in the *Bidirectional Cliffside Climbing* case.

3. How does the system determine whether a contact stage is fulfilled given a state at timestep $t$?

4. How does the author train the policy to fulfill the contact stages one by one? In the example of the *Dynamic Clap-and-Tap* case, the policy could easily learn to fulfill the similar contact stages first, then the different contact stages. How does the author prevent this from happening?

**Robotics Focus:**

4

**Summary Of Paper:**

This paper proposes a general framework of using contact positions and robot base pose as a high-level command representation to control the low-level humanoid robot policy. The author demonstrates the effectiveness of the proposed method in various case studies, including *Versatile Parkour Jumping*, *Dynamic Clap-and-Tap*, and *Bidirectional Cliffside Climbing*.

**Summary Of Recommendation:**

This paper provide valuable insight of how this contact sequences as command can be deployed in the real-world. But the technical details should be improved. I recommend a weak acceptance on this paper.

---

### Official Review · Reviewer_PP8V · 2024-07-21

**Originality:** 4
**Technical Quality:** 4
**Clarity Of Presentation:** 4
**Potential Impact:** 4
**Recommendation:** 4
**Confidence:** 5

**Review:**

Strengths:
- The paper makes a good novel proposition of what is currently missing from RL whole-body control: a “systematic and general method” for specifying the task, and proposes a good answer with the contact goal reward scheme that is shown to represent and facilitate optimization for diverse challenging tasks.
- The stage count rewards and curiosity rewards are task agnostic.
- The sim-to-real recipe seems good since it transfers diverse and dynamic behaviors to the real robot.

Weaknesses:
- The framework requires some manual regularizations for walking that need to be turned off when jumping. I'm curious if they can be replaced with WoCoCo rewards or there is a limitation in representing e.g. trotting interleaved with other contact based objectives.
- The weighting of the contact, stage count, curiosity reward, and task rewards still needs to be tuned for each task. So, while an impressive amount of the task spec is task agnostic, there is still room to further reduce the effort.

**Quality Of The Limitations Section:**

3

**Questions For Rebuttal:**

The framework does still require manual regularizations for walking that need to be turned off when jumping. Why not manually define a trotting or standing contact schedule for the feet during locomotion tasks (e.g.  https://arxiv.org/abs/2011.01387) and fully standardize the WoCoCo framework? Is there a limitation that things get tricky here, when the number of sequential contacts is large or unknown a priori, as with the foot contacts when walking to pick up a box? I guess you need to skip the stage reward and impose a manual timing for those trotting feet contacts..

Formulating the problem as sequential contacts might be limiting if the robot has grippers or hands. What you really want to know is not whether the hand is in contact but whether it is grasping, which is not an easy query to the simulator. Do you foresee any limitations of WoCoCo for embodiments with non-point end effectors?

**Robotics Focus:**

4

**Summary Of Paper:**

This paper aims to propose a unified framework for whole-body humanoid control tasks with sequential contacts. It describes a general reward recipe that can be applied to any task involving sequential contacts, and shows that it yields successful policies for four challenging humanoid tasks.

**Summary Of Recommendation:**

The paper proposes a novel formulation for humanoid whole-body control with sequential contacts, with impressive sim-to-real results. I have a few minor questions related to possible extensions.

---

### Author Rebuttal · Authors · 2024-08-09

Dear reviewers and AC,

Thank you for your constructive feedback! We have accordingly revised our paper (attached in the rebuttal file) based on the reviews. The changes are highlighted in blue.

Most of the changes are in the
1) Introduction section to improve clarity;
2) Limitation section;
3) Appendix to include more technical details.

Thank you again for your time and effort in reviewing our paper!

---

### Decision · Program_Chairs · 2024-09-04

**Decision:**

Accept

**Comment:**

The paper proposes WoCoCo, a unified framework for whole-body humanoid control tasks involving sequential contacts. The key idea is to use contact positions as high-level command representations to design task-agnostic reward functions for low-level control policies of humanoid robots. The paper received positive initial reviews from the three reviewers. The reviewers pointed out the work's strengths as 1) using contacts for reward formulation is general and applicable for diverse and challenging control tasks, and 2) the trained policies were successfully transferred from simulation to the real world. Meanwhile, they also pointed out its weaknesses as 1) manual regularization was required for walking but needs to be turned off for jumping, which limits the framework's flexibility, 2) manual efforts were required for tuning the weighting terms of reward components for individual tasks, and 3) writing clarify can be further improved, and some technical details should be further discussed.

**Post-rebuttal update:**

The AC appreciated the detailed responses in the authors' rebuttal. Toward the end of the discussion period, all three reviewers remained positive about this submission. Although some lingering (and minor) issues have been mentioned by Reviewer MQ1w and HtJA, which were not fully resolved by the rebuttal, the AC believes that this work has made sufficient technical contributions and demonstrated impressive behaviors on the humanoid robot. Therefore, the AC concurs with the reviewers and recommends accepting this paper for CoRL.